

# Hydrogen production by photocatalysis using new composites based on $SiO_2$ coated by $TiO_2$

Antonio Eduardo da H. Machado[1,2] and Werick Alves Machado[3]

[1] Instituto de Física, Universidade Federal de Catalão, Catalão, Goiás, Brazil
[2] Laboratório de Fotoquímica e Ciência de Materiais, Universidade Federal de Uberlândia, Uberlândia, MG, Brasil
[3] Instituto Federal de Educação, Ciência e Tecnologia do Triangulo Mineiro, Ituiutaba, Minas Gerais, Brazil

## ABSTRACT

In this study new $TiO_2$ photocatalysts core@shell type were synthesized using $SiO_2$ as structural support. The coating was confirmed by scanning electron microscopy and infrared spectroscopy. Adsorption isotherms revealed that the surface area of such composites is about 26% higher than pure oxide (W50). X-ray diffractograms combined with Raman spectroscopy revealed that the synthesized $TiO_2$ presents a structure based on the coexistence of anatase and brookite. The composite W50S50 presented the best photocatalytic performance of $H_2$ production, with 13.5 mmol in 5 h, corresponding to a specific rate of 32.5 mmol $h^{-1}g^{-1}$. In the reuse assays, this composite presented a good stability in the production of $H_2$. However, its performance presented a reduction of 23% over the reuse cycles. Considering the $H_2$ production in a solar simulator, W50S50 produced about 25.0 μmols, which is equivalent to 48.0 μmols $h^{-1}g^{-1}$, suggesting the good performance of this material for photocatalytic hydrogen production.

# INTRODUCTION

The rapid increase in global demand for energy has compromised the integrity of the environment with unprecedented speed. This is largely due to the current composition of the global energy matrix, predominantly composed of non-renewable sources such as oil, coal and natural gas (*International Energy Agency, 2018*). In view of this, new challenges in relation to energy generation and consumption have imposed to the society, influencing the research on energy generation, whose bias has changed towards the development of new renewable and non-polluting energy matrices. In this scenario, hydrogen ($H_2$) stands out as an alternative to the issues related to the energy matrix, since this gas can contribute to the production of a clean, safe and sustainable energy. However, the economically viable $H_2$ production still takes place through the exploitation of non-renewable sources. Currently, about 48% IEA of the global demand for this gas is met by steam reform of natural gas, 28% by oil reform, 20% by coal gasification, and 4% by other processes, such as electrolysis, biological and photocatalysis (*Levin & Chahine, 2010*). In the future, the $H_2$

Corresponding author
Werick Alves Machado,
werickalvez@hotmail.com

produced from environmentally friendly methods, such as by photocatalytic production *via* water splitting (*Veras et al., 2017*), may be an alternative way to circumvent the adverse effects of the use of non-renewable sources.

The photocatalytic production of $H_2$ *via* water splitting can occur by a well-established mechanism that involves the electronic excitation (photoactivation) of semiconductor materials (photocatalysts). The photoactivation promotes the formation of charged pairs, that is, positive (hole) and negative (electron) charge carriers, responsible for reduction and oxidation reactions. The pathway of oxidation leads to the formation of $O_2$, while the reduction to the formation of $H_2$ (*Bahnemann & Schneider, 2013*).

Among the most used materials, $TiO_2$ stands out due to characteristics such as abundance, low toxicity, good chemical stability in a wide range of pH, photocatalytic activity, photostability and insolubility in water (*Santos et al., 2015*; *Chong et al., 2010*; *Kandiel et al., 2010*; *Machado, Alves & Machado, 2019*). However, due to its relatively large band gap energy (Eg), situated in the near ultraviolet, its application in photocatalysis based on the use of solar radiation ends up being limited, since only 5% of the solar spectrum is in this region (*Kumar & Devi, 2011*). In addition, other limitations are found for the application of $TiO_2$ as a photocatalyst, such as the high recombination rate of the photogenerated charge carriers, morphology, aggregation of nanoparticles, *etc*.

In this sense, the present study proposes two simultaneous fronts to minimize trends that compromise $TiO_2$ performance, seeking to maximize the photocatalytic production of $H_2$: First, to promote structural modifications in $TiO_2$ with the creation of a biphasic (anatase/brookite) material. This mixture of phases tends to favor photocatalytic activity, minimizing the recombination of photoinduced charge carriers, since the cathode potential of the conduction band of brookite is approximately 0.14 eV lower than that of anatase (*Patrocinio et al., 2015*; *Machado & Machado, 2020*). Second, the immobilization of $TiO_2$ on silica ($SiO_2$) surface, giving rise to core@shell ($SiO_2@TiO_2$) type composites with different proportions of both species. The immobilization of $TiO_2$ on $SiO_2$ surface tends to increase the photocatalytic activity due to the lower tendency of $TiO_2$ to aggregate in this biphasic system (*Machado, Alves & Machado, 2019*), as well as the presence of oxygen vacancies in the supported $TiO_2$, which can contribute to minimize the recombination of photogenerated pairs (*Cheng et al., 2003*). Furthermore, Guo and coworkers (*Guo et al., 2014*) suggested that the use of silica as a support tends to increase the concentration of hydroxyl groups on the surface of the catalyst, making it more hydrophilic, which may favor the adsorption of reagents.

Thus, this study aims to obtain new photocatalysts based on $TiO_2$, with high performance in sustainable $H_2$ production, through the promotion of improvements in their photocatalytic parameters and reducing the aggregation of $TiO_2$ particles. In this sense, the performance of photocatalytic production of $H_2$ was evaluated using different biphasic composites involving $TiO_2$ supported in $SiO_2$. Different characterization techniques were used in order to relate the optical and morphological properties with the photocatalytic performance of these materials. Therefore, the present work intends to contribute with new insights into the positive effects of $TiO_2$ immobilization on the $SiO_2$

surface, aiming to potentiate the photocatalytic production of $H_2$ and its reuse in this noble application.

## MATERIALS AND METHODS

### Reagents

All reagents were of analytical grade, being used without prior treatment. Titanium tetraisopropoxide, 97%, isopropanol, 99.5%, hexachloroplatinic hexahydrate acid, 37.5%, tetraethyl orthosilicate, 98%, hydrochloric acid, 37% and sodium hydroxide, 98%, were supplied by Sigma-Aldrich (St. Louis, MO, USA). Methanol, 99.8%, was supplied by Dinâmica (Diadema, Brazil) and acetone 99.5% was provided by Synth (San Francisco, CA, USA). All solutions were prepared using ultrapure water obtained through an Elix 5 water purification system.

### Preparation of the photocatalysts

The standard photocatalyst, called W50, was obtained by the sol-gel method following a description done in a previous study (*Machado & Machado, 2020*). This process consists in the solubilization of titanium tetraisopropoxide in isopropanol at 3 °C under ultrasonic stirring for 20 min, followed by its hydrolysis by the addition, by dripping, of a water/acetone 50% v/v mixture, and precipitation under ultrasonic stirring. The resulting amorphous solid was washed with distilled water, centrifuged, being later sintered in conventional furnace at 400 °C for 5 h.

The composite $SiO_2@TiO_2$ was synthesized by the sol-gel method, coating $SiO_2$ nanoparticles with $TiO_2$. Prior to this synthesis, $SiO_2$ was prepared using the Stober method (*Stöber, Fink & Bohn, 1968*). In this method a mixture of 15 mL of deionized water, 4 mL of ammonium hydroxide, 100 mL of ethanol and 3 mL of tetraethyl orthosilicate are reacted at room temperature, under constant magnetic stirring for 1 h. Due the alkalinity of the resulting solution, it was subsequently neutralized with a solution 5.0 mol $L^{-1}$ of HCl. The resulting solid, $SiO_2$, was washed using deionized water, centrifuged and dried.

The coating of $SiO_2$ by $TiO_2$ involved the previous dispersion of silica, in the estimated amount by stoichiometric calculation, in 150 mL of 2-propanol under magnetic stirring for 1 h, followed by the rapid addition of 10 mL of titanium isopropoxide to the suspension. This mixture was maintained under vigorous magnetic stirring for 19 h. Subsequently, in the hydrolysis of the titanium precursor, a water/acetone (50% v/v) mixture was added drop by drop to the mixture, which was kept under magnetic stirring for 1 h. Finally, the resulting colloidal suspension was centrifuged, being the precipitate separated and submitted to the same heat treatment provided to the photocatalyst used as reference (*Machado & Machado, 2020*).

The composites were synthesized using three different proportions of $SiO_2$ (25%, 50%, 75% m/m) in relation to $TiO_2$, being named W50S25, W50S50 and W50S75, respectively.

### Characterization of photocatalysts

The composites were characterized by different techniques:

By infrared spectroscopy (FTIR), using a Perkin Elmer MIR Frontier Single spectrometer. The analysis of the samples was performed in the solid state, in the region between 4,000 and 220 cm$^{-1}$, and resolution of 4 cm$^{-1}$, using an Attenuated Total Reflectance (ATR) accessory.

The scanning electron microscopy (SEM) images and the EDS spectra were obtained using a Tescan Vega 3 electronic microscope equipped with a secondary electron detector, with an acceleration voltage of 5.0 kV. From the images obtained by SEM and with the help of the *ImageJ* software, it was possible to calculate the particle size by randomly selecting approximately 100 particles per image. From there, the histograms were built for the synthesized oxides, which illustrate the average particle size distribution.

The surface area, porosity and pore volume measurements were performed from the analysis of adsorption and desorption isotherms of $N_2$, using Quantachrome equipment, model NOVA touch LX1. In these assays, the samples were pretreated under flow of gaseous $N_2$ for 12 h, at 120 °C, in order to remove adsorbed gases and water. The measures were done at 77 K using liquid $N_2$ to maintain the temperature during the analyses. The surface areas were estimated using the method proposed by *Brunauer, Emmett & Teller (1938)* (BET) to analyze the adsorption data, while the method proposed by *Barrett, Joyner & Halenda (1951)* (BJH) was used to calculate the pore volume.

The X-ray diffractograms were obtained using a Shimadzu XRD-6000 diffractometer (Shimadzu, Kyoto, Japan), equipped with a CuKα ($\lambda$ = 1.54148 nm) monochromatic font, in the 10° ≤ 2θ ≤ 80° angular range. The counting step was 0.02° and scanning speed of 0.5°/min. Finally, the spectra were refined by Rietveld's method using the software FullProf (*Roisnel & Rodriguez-Carvajal, 2001*). As a criterion of reliability and quality of refinement, the obtained S factor was less than 1.37 for all photocatalysts.

The Raman spectra were obtained using a Horiba LabRAM HR Evolution spectrometer (Horiba, Kyoto, Japan) with 600 lines/mm grid, equipped with an excitation laser at 633 nm, with power of 5 mW. The spectra obtained were the result of the accumulation of eight scans in the range between 100 and 1,000 cm$^{-1}$.

The optical absorption spectra in the diffuse reflectance mode were obtained using a Shimadzu UV-1650 spectrophotometer coupled to an integrating sphere, using barium sulfate as standard. These spectra were obtained at room temperature in the spectral range between 200 and 800 nm, being converted in terms of Kubelka-Munk's function (F(R)), thus being possible to directly estimate the *band gap* (Eg) of the studied materials (*Liu & Li, 2012*).

## Photocatalytic production of H$_2$

This was evaluated by three different approaches. In the first, the most active composite was identified among the synthesized materials, which was done on bench scale experiments, monitoring the production of H$_2$ achieved in the same time interval by each photocatalyst. In the second, also on bench scale, the most efficient composite was submitted to reuse assays, in a process involving four consecutive photocatalytic cycles. Finally, in a solar simulator the production mediated by the most efficient composite and pure oxide (W50), was evaluated.

The photocatalytic system used in bench scale (*Machado & Machado, 2020*) is based in a reactor built in borosilicate glass with total volume of 1.5 L, possessing a cooling jacket also made in borosilicate glass. It is connected to a thermostatic bath, which keeps the reaction medium at 20 °C. The reactor was positioned on a magnetic agitator, used to promote homogenization of the aqueous suspension containing the catalyst and reactive species. A 400 W high pressure (HPL-N) mercury lamp without its protective bulb was used as radiation font. The photonic flow of this lamp was estimated to be $3.3 \times 10^{-6}$ Einstein/s (*Machado et al., 2008*), and irradiance equal to 100 W/m² in the UVA. The lamp was positioned laterally at 15 cm from the reactor.

Before the assays of $H_2$ production the photocatalysts were loaded by photoreduction with 0.1% m/m of Pt, obtained from a solution of hexachloroplatinic acid. The photocatalyst loaded with Pt was suspended in 750 mL of a water/methanol mixture containing 20% v/v of methanol, used as a sacrificial reagent. The pH of the reaction medium was adjusted to 6.2, isoelectric point of $TiO_2$, pH at which its photocatalytic activity is favored according to studies by *Hoffmann et al. (1995)*. For this, 0.1 mol $L^{-1}$ solutions of HCl and NaOH were used for adjustment. Before each experiment, the dissolved gases, especially oxygen, inside the reactor were purged with $N_2$ for 5 min. Finally, with the lamp on, the photocatalytic assays were started. During the reaction, aliquots of the gases produced were collected every hour, in a total period of 5 h. The gases were analyzed by gas phase chromatography using a PerkinElmer Clarus 580 chromatograph (PerkinElmer, Watham, MA, USA), equipped with two packed columns (porapak N 2 mm and molecular sieve) and a thermal conductivity detector (TCD). All experiments were carried out at least in triplicate.

In the assays of $H_2$ production using a solar simulator a smaller volume reactor was used. All experimental parameters such as reaction time, initial pH, photocatalyst concentration, sacrifice reagent and cocatalyst were maintained proportionally equal to those employed on bench scale, for comparative purposes. The solar simulator, described by *Nunes, Patrocinio & Bahnemann (2019)*, is constituted by a reactor, also made of borosilicate glass, with an internal volume of 80 mL, a 300 W xenon lamp, used as radiation font, and an AM1.5 filter, which simulates solar conditions after radiation passes through 1.5 times the atmospheric mass. This is equivalent to the direct incidence of solar radiation on earth's surface, with a deviation of 48.2° from the angle of zenith (*Honsber & Bowden, 2019*). The reactor cooling system was connected to a thermostatic bath to keep the reaction medium at 20 °C. During the reactions, the content inside the reactor was maintained under stirring. The reactor was positioned at 15 cm from the radiation source, being exposed to an irradiance of 20 W/m² in the UVA.

For comparative purposes, since different photocatalytic systems were used, in addition to the amount, in mols, of produced $H_2$, the results were expressed in terms of specific rate of $H_2$ production (SRHP) (*Machado, Alves & Machado, 2019*; *Lin & Shih, 2016*),

$$SRPH = \frac{n}{t \, m}$$

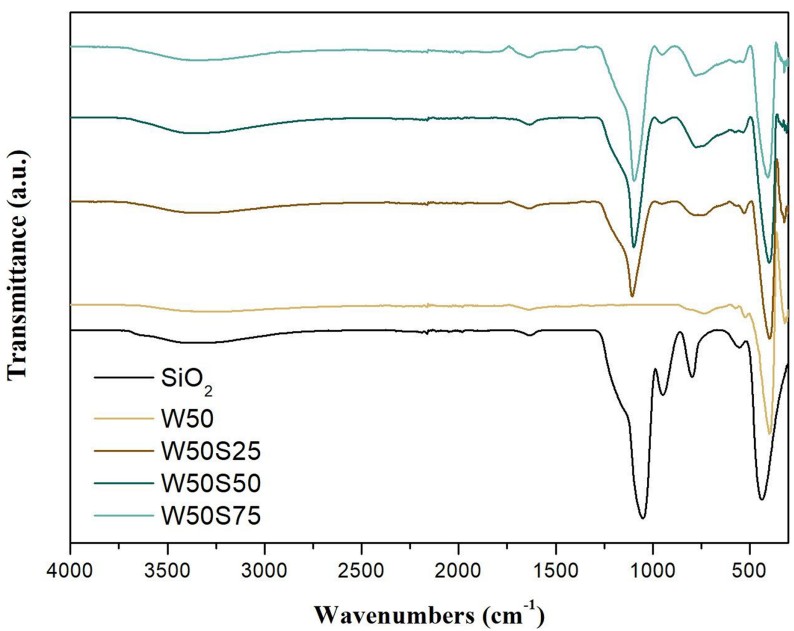

**Figure 1 Infrared transmission spectra obtained for the synthesized oxides.**

where $n$ is the number of mols of $H_2$, obtained by integration in the interval between 4 and 5 h; $t$ is the time of reaction; $m$ is the mass of catalyst (g).

# RESULTS AND DISCUSSION

## Characterizations

Figure 1 presents the FTIR spectra of pure $TiO_2$, $SiO_2$, and of the $TiO_2/SiO_2$ composites (W50S25, W50S50 and W50S75). It is possible to distinguish three typical vibrations related to pure $SiO_2$: bands at 438 and 803 cm$^{-1}$ and a band centered at 1,050 cm$^{-1}$ related, respectively, to bending and symmetrical and asymmetric stretching of Si-O-Si, and a secondary vibration at 960 cm$^{-1}$ related to silanol groups (Si-OH) (*Panwar, Jassal & Agrawal, 2016*; *Kermadi et al., 2015*). For pure $TiO_2$, three characteristic bands are observed: a wide and intense at 403 cm$^{-1}$ and two more subtle, at 530 and 730 cm$^{-1}$, both related to Ti-O-Ti stretching (*Mohamed, Osman & Khairou, 2015*). The composites present the two most intense bands of the respective oxides, with small displacements: a band centered at 1,102 cm$^{-1}$, associated to silica, and a band at 403 cm$^{-1}$, related to $TiO_2$.

The low signal intensity referring to the silanol groups in the composites proves the silica coating by $TiO_2$. These groups, present on the silica surface, assists in the stability of metal charges through the Si-O-M bonds, favoring the dispersion of $TiO_2$ over the support (*Almeida et al., 2004*; *Chen et al., 2018*). It is also observed that composites with higher concentrations of $SiO_2$ have a more intense band centered at 960 cm$^{-1}$ due to silica not covered by $TiO_2$.

The Si-O-Si vibration, in general at 1,050 cm$^{-1}$, is slightly shifted to higher frequencies (approximately 1,102 cm$^{-1}$) due to calcination of these materials at high temperature, suggesting the strengthening of the Ti-O bond (*Kermadi et al., 2015*). In general, in all

**Table 1  Morphological parameters related to SiO$_2$, TiO$_2$ and composites.**

| Photocatalyst | Average particle size (μm) | Surface area (m$^2$/g) | Porosity (%) | Mean pore diameter (nm) |
| --- | --- | --- | --- | --- |
| W50 | 0.7–1.0 | 103 | 16.5 | 6.0 |
| W50S25 | 0.5 | 130 | 19.5 | 6.0 |
| W50S50 | 0.4 | 120 | 21.0 | 7.0 |
| W50S75 | 0.4 | 113 | 20.5 | 8.0 |
| SiO$_2$ | 0.2 | 42.0 | 5.0 | 4.0 |

spectra it was possible to observe the presence of two relatively wide bands at 1,636 and 3,360 cm$^{-1}$, related to O-H vibration of water molecules chemically and physically adsorbed on the surface of the photocatalyst (*Poo-Arporn et al., 2019*).

In Figs. S1–S5, the images obtained by SEM are shown. They are accompanied by their EDS spectra and histograms, that illustrate the average particle size distribution of photocatalysts (Table 1). From the analysis of the figures, it is observed that the particles of pure TiO$_2$ have a dense aspect, with irregular spherical shape and average particle sizes ranging from 0.2 to 1.0 μm. Pure silica presents particles of regular spherical shape, with uniform average size. However, for them there is a tendency to aggregation, giving rise to bulky clusters of SiO$_2$. For the studied composites, when compared to the W50, the reduction of the average particle size between them is evident, and this occurs as consequence of the coating of SiO$_2$ by TiO$_2$ (*Li et al., 2013*). For composite W50S75 it is verified through the analysis of the histogram, the existence of excess of SiO$_2$, with the presence of particles with average diameter of 0.2 μm, in addition to a considerable increase in the aggregation state.

The isotherms, Fig. S6, suggest that the photocatalysts under study are type IV, characterized by being mesoporous, with average pore diameter between 2 and 50 nm, corroborating with the average pore diameter values, as displayed in Table 1. The profile of hysteresis are similar to type H$_2$, which correspond to complex mesoporous structures, in which the distribution of pore size and shape is not well defined. Silica, in turn, presented type I isotherm and hysteresis characteristics of microporous materials, composed by agglomerates of spheroidal particles with close size distribution (*IUPAC, 1985*). Due to this, it is observed that the silica hysteresis loop does not close a typical behavior of materials with very narrow pores or bottle-shaped pores. This evidences the low average pore size, that prevents the diffusion of adsorbed N$_2$ (*Tang et al., 2017*). The observed porosity also corroborates with the formation of the composite since the immobilization of TiO$_2$ on the silica surface leads to the formation of new pores, resulting in increased porosity and consequent increase in the surface area (*Salgado & Valentini, 2015*). The composite W50S50 presented the highest porosity among the synthesized composites, which ensures greater adsorption of reagents on its surface, consequently increasing its photocatalytic action.

By analyzing the diffractograms shown in Fig. 2, and confronting with information reported in literature (*Machado & Machado, 2020*; *Neto et al., 2017*) and with the JCPDS

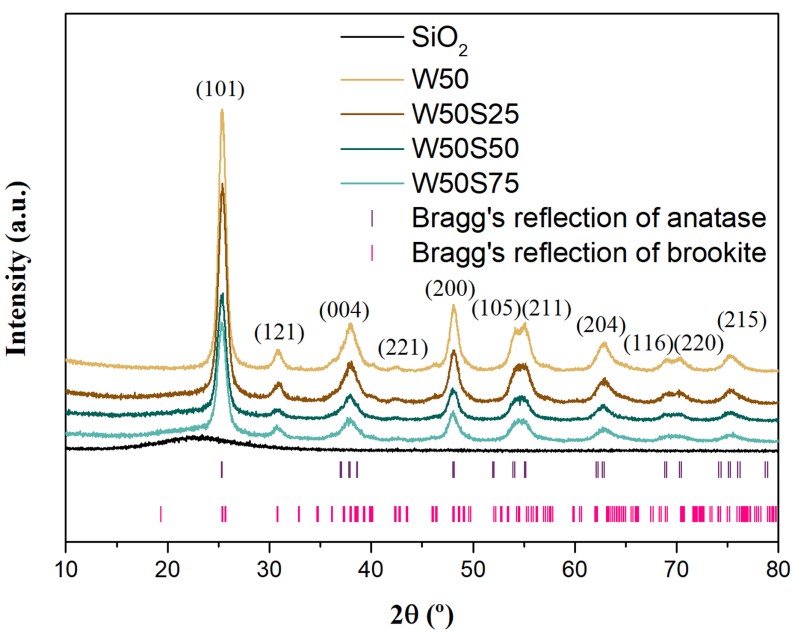

**Figure 2  X-ray diffractogram for the synthesized oxides.**

21-1272 (anatase) and 29-1360 (brookite) crystallographic files, it is possible to observe the coexistence of two crystalline phases in all the synthesized oxides, being anatase the main phase and brookite the secondary. It is worth mentioning the similarity of the diffraction profiles of these materials, with emphasis on the well-defined diffraction peaks for the uncoated material (W50), suggesting a significant increase in the crystallinity as a consequence of the heat treatment applied during the synthesis (*El-Sheikh et al., 2017*). On the other hand, it is observed that the intensity of the peaks decreases as the concentration of silica increases, indicating a decrease in the crystallinity of the composites, consequence of the non-crystallinity of $SiO_2$ (*Machado, Alves & Machado, 2019*).

From the diffractograms, it was possible to obtain, through Rietveld refinement, the proportion of crystalline phases, average size and the average deformation of the crystallite for synthesized species, Table 2. The diffractograms, accompanied by their respective calculated diffraction profiles, experimentally obtained profile, residual curves and Bragg diffraction adjusted by the Rietveld method, can be viewed in Fig. S7. The reliability factors of refinement are shown in Table 1.

The presence of brookite phase in the composites was verified in the oxide designated as W50, synthesized in a previous study (*Machado & Machado, 2020*). This involved the use of water/acetone mixtures to control the hydrolysis of titanium tetraisopropoxide. The presence of acetone during the synthesis affects the organization of the critical nuclei formed from the oligomeric network generated from titanates, which tends to favor the formation of the brookite phase (*Machado & Machado, 2020*). On the other hand, the addition of silica during the synthesis of the materials evaluated in the present study apparently did not influence the formation of any crystalline phase.

**Table 2 Percentage of crystalline phase, size and average maximum crystallite deformation, obtained from Rietveld refinement, for TiO$_2$ and composites.**

| Photocatalyst | Crystal phase (%) | Mean size of crystallite (nm) | Average maximum crystallite deformation (%) |
|---|---|---|---|
| W50 | Anatase 76.0 | 32 | 9 |
| | Brookite 24.0 | 14 | 10 |
| W50S25 | Anatase 75.5 | 27 | 12 |
| | Brookite 25.5 | 15 | 12 |
| W50S50 | Anatase 70.0 | 22 | 13 |
| | Brookite 30.0 | 9 | 19 |
| W50S75 | Anatase 73.0 | 6 | 15 |
| | Brookite 27.0 | 10 | 23 |

In the present study, the composite W50S50 presented the highest percentage of brookite among the synthesized materials—an increase of 25% compared to pure oxide. As shown in Table 2, the other composites showed similar proportions between the crystalline phases. The average crystalline size of the composites for both crystalline phases decreased with increasing silica concentration in the structure. A more expressive reduction was observed for composite W50S75, which presented contraction of the anatase phase greater than five times in comparison with pure oxide. This suggests that the presence of silica should inhibit the growth and surface diffusion processes of TiO$_2$ nanoparticles due the curvature of the silica surface and the formation of interfacial bonds between oxides (*Machado, Alves & Machado, 2019*; *Li et al., 2013*). It was also found that the behavior of the mean maximum deformation was inversely proportional to the average crystallite size, since the formation of interfacial bonds tends to compromise the integrity of the anatase and brookite crystals. According to Staykov (*Staykov et al., 2017*), the strong Si-O-Ti bonds at the composite interface tensions the crystalline network of TiO$_2$ and can change the coordination sphere of Ti$^{4+}$ from six to five coordinated O$^{2-}$. Thus, the increase in silica concentration in the TiO$_2$ structure leads to an increase in tension in the crystalline network, which tends to increase the average deformation of the material (*Staykov et al., 2017*).

As well as the diffractograms obtained by X-ray diffraction, the Raman spectra shown in Fig. 3 also evidence the biphasic composition of these photocatalysts. The active modes corresponding to the anatase phase are located at 144 cm$^{-1}$ (E$_g$), 197 cm$^{-1}$ (E$_g$), 399 cm$^{-1}$ (B$_{1g}$), 513 cm$^{-1}$ (A$_{1g}$), 519 cm$^{-1}$ (B$_{1g}$) and 639 cm$^{-1}$ (E$_g$) (*Staykov et al., 2017*; *Sekiya et al., 2001*). For the analyzed samples, five of these main bands are observed in the following regions: 144 cm$^{-1}$ (E$_g$), 198 cm$^{-1}$ (E$_g$), 399 cm$^{-1}$ (B$_{1g}$), 519 cm$^{-1}$ (B$_{1g}$) and 640 cm$^{-1}$ (E$_g$). The mode A$_{1g}$ at 513 cm$^{-1}$ was probably not visualized due to its low intensity in combination with the overlay of the mode B$_{1g}$, more intense and close to 519 cm$^{-1}$ (*Iliev, Hadjiev & Litvinchuck, 2013*; *Fang et al., 2015*). In the insert in Fig. 3, between 200 and 500 cm$^{-1}$ four subtle bands attributed to the brookite phase are observed in 245 cm$^{-1}$ (A$_{1g}$), 321 cm$^{-1}$ (B$_{1g}$), 365 cm$^{-1}$ (B$_{2g}$) and 452 cm$^{-1}$ (B$_{3g}$). The band of higher intensity for

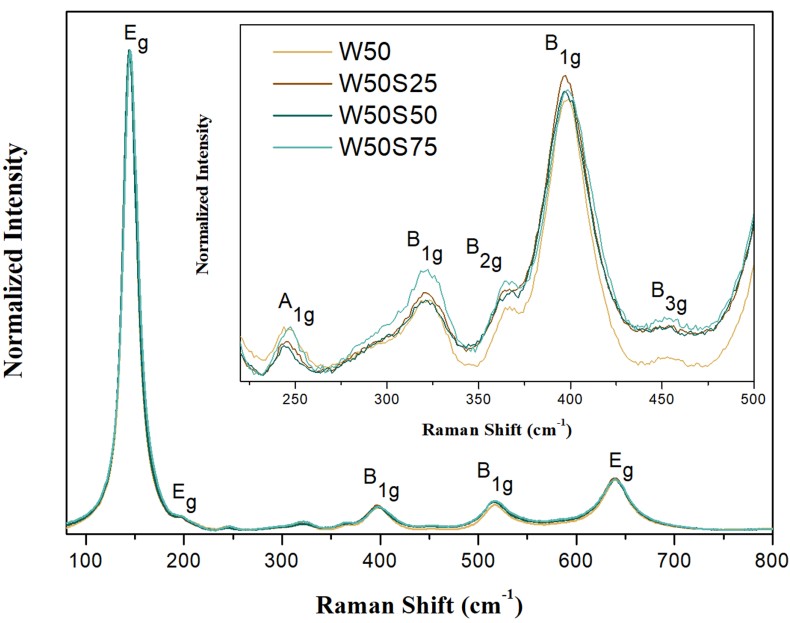

**Figure 3 Raman spectra of the synthesized materials: pure TiO$_2$ and composites.** Insert: bands at 245, 321, 365 and 452 cm$^{-1}$, attributed to brookite phase.

the phase brookite is close to 153 cm$^{-1}$ (A$_{1g}$). It may be overlapthed with the anatase band, much more intense, at 144 cm$^{-1}$ (E$_g$), which should influence the wider width of this Raman mode (*Yin et al., 2007*; *Hellawell et al., 2015*). In Fig. S8, the Raman spectrum of pure silica is shown, in the spectrum well-defined Raman bands between 200 and 600 cm$^{-1}$ are not observed, but it has a single wide band between 1,000 and 2,250 cm$^{-1}$, characteristic of tetrahedrons SiO$_4$ jitters typical of amorphous SiO$_2$ as shown by X-ray diffraction (*Sahoo, Arora & Sridharan, 2009*).

The estimated values for Eg from the diffuse reflectance spectra, Fig. 4, are 3.25 eV for W50, 3.32 eV for W50S25 and W50S50, and 3.35 eV for W50S75. These results agree with the reported in the literature, which suggests an Eg of approximately 3.2 eV for pure TiO$_2$ (*Neto et al., 2017*). As observed above, the estimated value for E$_g$ for the synthetized composites is slightly higher, probably due the formation of interfacial bonds (Si-O-Ti), which tends to change the electronic frontier states. In addition, it is known that the electronic properties of particles tend to change significantly with the reduction of their size due the occurrence of quantum confinement, which tends to increase the Eg value (*Kumar & Devi, 2011*). As the results obtained by TEM and XRD suggest, composites that have higher silica content in their composition have smaller particulate and crystalline sizes and consequently higher Eg values. It is worth noting that amorphous silica possesses Eg higher than 8.0 eV (*Machado, Alves & Machado, 2019*; *Nekrashevich & Gritsenko, 2014*).

## Photocatalytic production of H$_2$ in a bench-scale

Being known the morphological and optical properties of the materials presented in this work, the photocatalytic activity regarding the photocatalytic production of hydrogen was

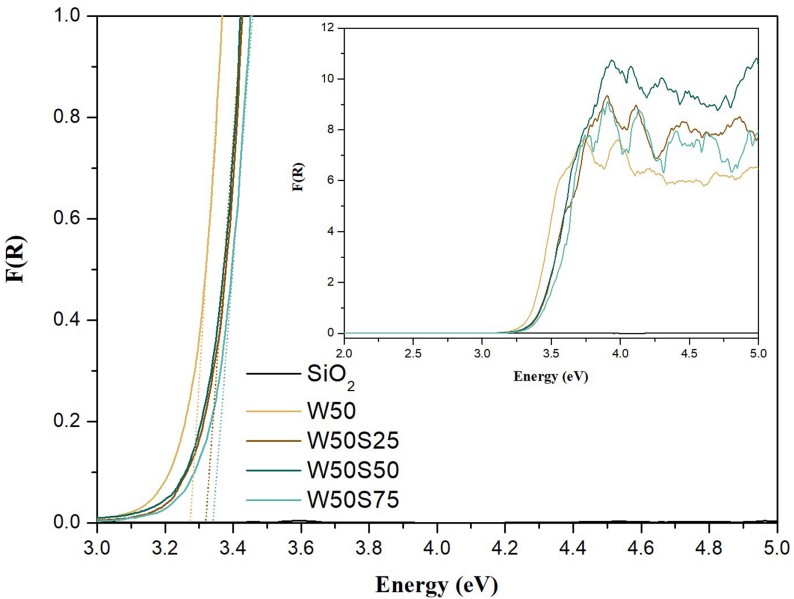

**Figure 4 Diffuse reflectance spectra of SiO₂ and composites in an enlarged scale *vs* Kubelka-Munk's function, combined to linear segments used in determining E_g values.** Insert: diffuse reflectance spectra, without magnification, for the synthesized materials.

evaluated through bench-scale tests under UV-vis irradiation. After defining the most efficient photocatalyst, its potential for reuse was evaluated. Following, the performance of these same composites in the production of $H_2$, compared to that of pure oxide, was evaluated using a solar simulator.

It is observed, from Fig. 5, that the synthesized composites obtained better performance in the production of $H_2$ than the pure oxide (W50), with the exception of W50S75, responsible for the lowest productivity of the set, about 20% less than that achieved using the W50. As observed in SEM images along with histograms and EDS spectra, the poor performance of the W50S75 should be attributed to excess of free silica, photocatalytically inert, in its composition. The most efficient composite, W50S50, produced approximately 13.5 mmols of $H_2$ in 5 h of reaction, performance 40% higher than that obtained using the W50, which produced about 9.6 mmols in the same time interval. Using the W50S25, a production of 11.0 mmols was achieved, a value approximately 14% higher than that obtained using pure oxide. It is noteworthy that $H_2$ production tests were also performed using pure silica associated with platinum and platinum powder. However, as expected, no $H_2$ production was obtained in both situations, in 5 h of reaction.

The more efficient production of $H_2$ by W50S50 was favored by the higher anatase/brookite heterophasic crystalline composition, possibly due the more negative cathode potential of the conduction band of the brookite phase. This tends to favor the reduction of protons during the production of $H_2$ (*Patrocinio et al., 2015*; *Machado & Machado, 2020*; *Tay et al., 2013*). Morphological aspects, such as the high surface area, high porosity and lower particle size, should also contribute to potentiate the photocatalytic activity of this oxide.

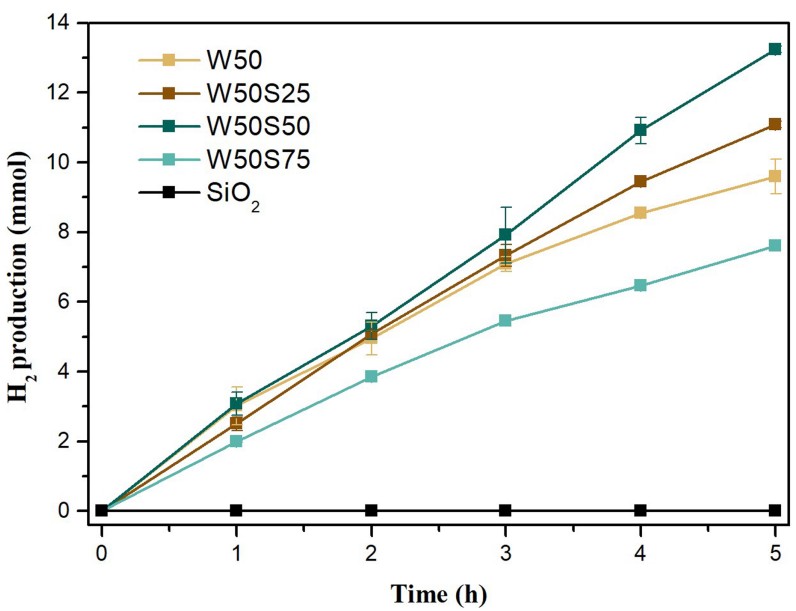

**Figure 5 Photocatalytic production of hydrogen *vs* reaction time for SiO$_2$ and composites.**

In a previous study published by us (*Machado, Alves & Machado, 2019*) 5.5 mmols of H$_2$ were produced in 5 h of reaction, which is equivalent to a SRHP of 13.6 mmol g$^{-1}$ h$^{-1}$, using a TiO$_2$/SiO$_2$ composite based on TiO$_2$ 100% anatase, and approximate composition of 80% of TiO$_2$ and 20% of SiO$_2$. The increase in photocatalytic performance in terms of H$_2$ production using the biphasic composite W50S50 is indisputable since, using these same experimental conditions, it was achieved, in the present study, a SRHP approximately two and a half times larger. Regarding a comparison of the photocatalytic action of W50S50 and the other catalysts evaluated in the present study with other catalysts reported in the literature, developed for the same purpose, it is worth mentioning the work of Lin and coworkers (*Lin, Yang & Wang, 2011*) who evaluated the efficiency of Nb$_2$O$_5$ combined with different metals as cocatalyst. They used a halogen lamp of 400 W as radiation font and aqueous solutions containing 20% of methanol. Under the best conditions, using platinum as cocatalyst, a SRHP of 4.6 mmol h$^{-1}$g$^{-1}$ was reached. In the present study, using W50S50 as catalyst, a SRHP seven times higher was achieved. In a study involving the photodeposition of CuO on the surface of ZnO, Liu and coworkers obtained, under the best conditions, a SRHP of 1.7 mmol h$^{-1}$g$^{-1}$ (*Liu et al., 2011*), using the catalyst suspended in aqueous solutions containing 10% v/v of methanol and irradiated by a 400 W mercury vapor lamp. Also, using the composite W50S50 as catalyst, we obtained a much higher SRHP.

## Potential for reuse of W50S50 for H$_2$ production in a bench-scale

The evaluation of the potential of reuse of the photocatalyst consisted of measuring the reproducibility of catalytic action of the W50S50 by repetitive tests, called cycles, using the same initial conditions applied to the system, only with the pH of the medium being

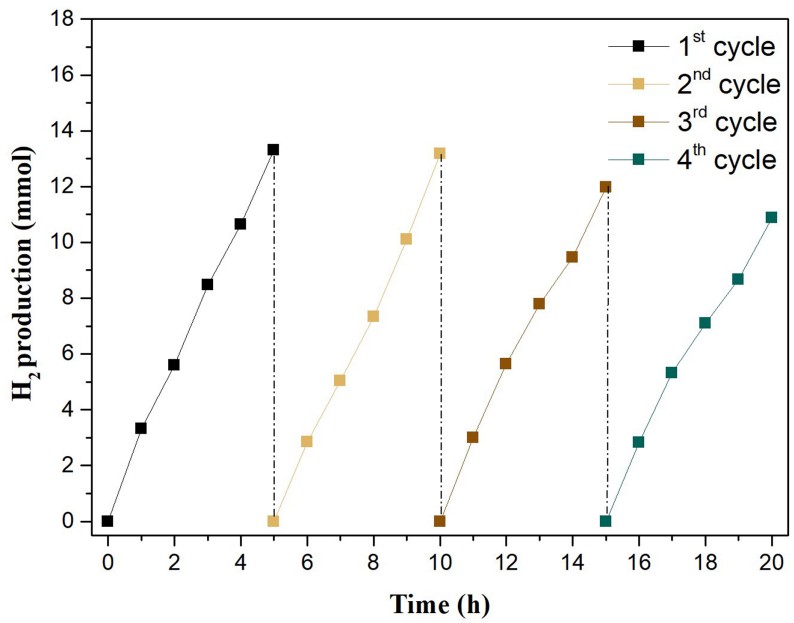

**Figure 6 Amount of H$_2$ produced by the photocatalyst W50S50 in four cycles of reuse.**

**Table 3 Amount, in mmol of H$_2$, produced, specific rate of H$_2$ production (SRHP), initial and final pHs for each cycle in the reuse assay.**

| Cycle | $n$ (mmols) | SRHP (mmol h$^{-1}$g$^{-1}$) | Initial pH | Final pH |
|-------|-------------|------------------------------|------------|----------|
| 1$^{st}$ | 13.5 | 32.0 | 6.17 | 5.78 |
| 2$^{nd}$ | 13.0 | 31.0 | 6.20 | 5.38 |
| 3$^{rd}$ | 12.0 | 28.5 | 6.15 | 4.45 |
| 4$^{th}$ | 11.0 | 26.0 | 6.15 | 4.30 |

adjusted at the beginning of each additional cycle. The reuse assays were performed in sequence of four photocatalytic cycles, each involving 5 h of reaction. The first cycle was equivalent to the H$_2$ production test carried out in a single cycle of 5 h.

Figure 6 shows the very good stability in H$_2$ production in each cycle, which maintains, individually per cycle, a regular upward pattern, within 20 h of the tests. However, there is a subtle decrease in production in each cycle when compared to the previous cycle.

Table 3, that contains the compilation of Fig. 6 data, shows the number of mols of H$_2$ produced in each cycle of reuse together with the respective SRHP. There was a reduction of approximately 18% in the production of H$_2$ between the first and last cycle. The SRHP of 32.0 mmol h$^{-1}$g$^{-1}$ reached in the first cycle was reduced to 31.0 mmol h$^{-1}$g$^{-1}$ in the second cycle, an almost equivalent value, whereas in the last cycle this value was reduced to 26.0 mmol h$^{-1}$g$^{-1}$. This can be attributed to the exhaustion of the sacrificial reagent stock. Methanol, when oxidized, tends to become formaldehyde and formic acid (*Mcmurry, 2011*). The pH at the end of each cycle was monitored in order to qualitatively verify the formation of formic acid during the reaction. In fact, it was observed that the pH was

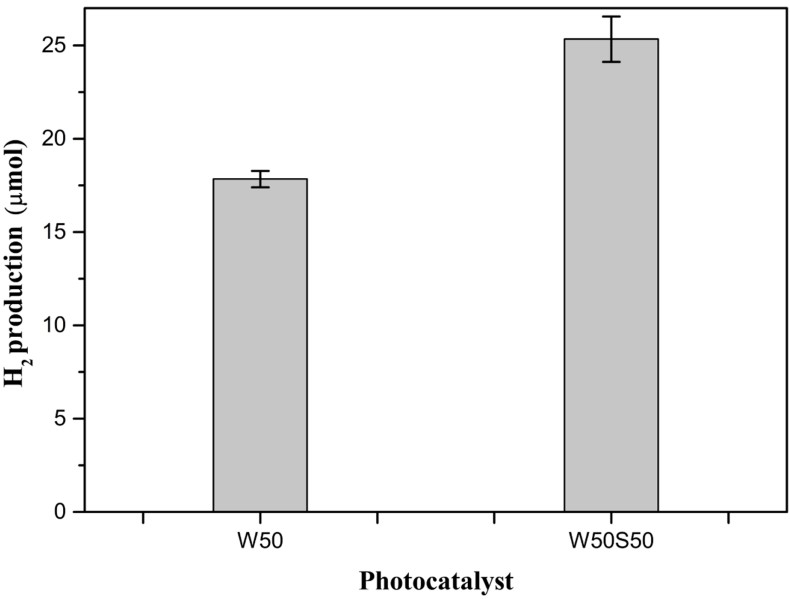

**Figure 7 Amount of $H_2$, in µmols, produced by the photocatalysts W50 and W50S50, using a solar simulator.**

reduced during the reuse cycles. From the third cycle, a significant reduction in the pH as well as of SRHP is evident.

Figure 7 shows the amount of H produced by each photocatalyst using the solar simulator, accompanied by their respective standard deviations. It is observed that the compound W50S50 maintains a high performance in comparison with pure oxide. Similar to that observed in bench-scale production, the production using this composite is also approximately 40% higher. Despite the electronic absorption by this material is subtly shifted to the region of higher energies (Fig. 4), its production in 5 h of reaction was 25 µmols against 18 µmols using the pure oxide, which is equivalent to, respectively, SRHPs of 66.6 and 48.0 $µmol\ g^{-1}h^{-1}$. Thus, it is clear that the synergism between $SiO_2$ and $TiO_2$ tends to favor the photocatalytic activity in the production of $H_2$. Even with the subtle hypsochromic displacement observed, in the mediation of the photocatalytic process by the solar simulator using the composite, the result was superior than that achieved using pure oxide.

Due the amount of $H_2$ produced using the solar simulator and the detection limit of the gas chromatograph, in this experiment it was not possible to monitor the temporal evolution of $H_2$. In this case, only aliquots collected with 5 h of reaction were analyzed, and each experiment was replicated at least three times.

## CONCLUSIONS

Infrared and dispersive energy spectra in addition to scanning electron microscopy images confirmed the coating of $SiO_2$ nanoparticles by $TiO_2$. The images obtained by scanning electronic microscopy also revealed that the spherical shape of the composite nanoparticles present high regularity most likely due the immobilization of $TiO_2$ on Stober' sílica surface. It is also worth mentioning that the average size of the composite nanoparticles was

uniform with a slight reduction compared to the size of the pure oxides that constitute them.

The $N_2$ adsorption and desorption isotherms demonstrated that the synthesized composites are mesoporous materials with mean pore sizes between 3 and 4 nm with approximately 20% of porosity, but without defined distribution and form. The surface area of these composites, calculated by employing the BET method, is approximately 26% higher in relation to pure oxide (W50).

The difratograms, together with the Raman spectra, revealed crystalline materials with the coexistence of anatase, as the main phase, and brookite. In addition, the diffractograms refined by the Rietveld method demonstrated that the composites maintained a proportion of about 75% and 25%, respectively of anatase and brookite. The average size of the crystallites has changed due the synthesis. In both crystalline phases there was a reduction in the average size with the increase of silica concentration in the structure. This suggests that the presence of silica inhibits the growth and surface diffusion processes of $TiO_2$ nanoparticles, probably due the curvature of the silica surface and formation of interfacial bonds between oxides.

The *band gap* energies estimated for the composites were slightly higher than that shown by standard oxide (3.2 eV). This should possibly be related to the mixture of electronic states of both materials ($SiO_2@TiO_2$), since Stober's amorphous silica presents an Eg higher than 8.0 eV.

In the photocatalytic assays of $H_2$ production on bench scale, the composites, in general, showed excellent photocatalytic performance, probably due the lower mobility of $TiO_2$ in view of its fixation to silica surface. The composite W50S50 proved to be the most efficient, since with 5 h of reaction it was achieved the production of approximately 13.5 mmol of $H_2$, a value 40% higher than that achieved using pure $TiO_2$. The good performance of this composite should be related to its morphological parameters, such as its high surface area, crystallite size, particle size smaller than other composites, and mainly to its higher heterophasic crystalline composition, since the cathode potential of brookite phase conduction band is more negative, favoring the reduction of protons in the production of $H_2$. Silica, which does not present photocatalytic activity, did not produce $H_2$.

In the reuse assays of W50S50, it showed excellent stability during the production of $H_2$. However, there was a decrease of 23% between the first and the last cycle, attributed to the exhaustion of the stock of sacrificial reagent.

In the photocatalytic production of $H_2$ using a solar simulator, the performance of W50S50 remained 40% superior to that obtained using the pure oxide. In 5 h of reaction, this composite produced 25 μmols against 18 μmols using the pure oxide, which equates to a SRHP of, respectively, 66.6 and 48.0 μmol $g^{-1}h^{-1}$.

## ACKNOWLEDGEMENTS

The authors thank the Institute of Physics of Uberlandia Federal University (UFU) for the RAMAN spectrum measures, the Research Group in Inorganic Materials (UFU) for the BET measures, and the Institute of Chemistry of Uberlandia Federal University for the scanning electron microscopy images.

### Funding
This work was supported by the Brazilian agencies CAPES, CNPq and FAPEMIG. The funders had no role in study design, data collection and analysis, decision to publish, or preparation of the manuscript.

### Grant Disclosures
The following grant information was disclosed by the authors:
Brazilian agencies CAPES, CNPq and FAPEMIG.

### Competing Interests
The authors declare that they have no competing interests.

### Author Contributions
- Antonio Eduardo da H. Machado conceived and designed the experiments, analyzed the data, authored or reviewed drafts of the article, and approved the final draft.
- Werick Alves Machado conceived and designed the experiments, performed the experiments, analyzed the data, prepared figures and/or tables, authored or reviewed drafts of the article, and approved the final draft.

### Data Availability
   The raw measurements are available in the Supplemental Files.

### Supplemental Information
Supplemental information for this article can be found online at http://dx.doi.org/10.7717/peerj-matsci.25#supplemental-information.

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
