# Peer review of "Hydrogen production by photocatalysis using new composites based on $SiO_2$ coated by $TiO_2$"

_PeerJ Materials Science, doi:10.7717/peerj-matsci.25_

## Round 0.1 · original submission · Major Revisions

The reviewers identified a series of corrections/additions needed before the manuscript can be accepted. Please consider each of the suggestions and address them while resubmitting. I am confident that this will help enhance the manuscript's quality.

Reviewer 1 ·

Basic reporting

The authors have reported new TiO2 photocatalysts core@shell type were synthesized using SiO2 as
structural support. They have provided appropriate literature reviews and background details of their work. Provided high quality publishable figures in the main manuscript as well as in supplementary information.

Experimental design

The authors have provided elaborate details regarding their experimental design and adopted Appropriate methodologies for the work (scanning electron microscopy, infrared spectroscopy, Raman spectra etc).

Validity of the findings

Based on their experimental approach, the authors have demonstrated that the spherical shape of the composite nanoparticles present high regularity most likely due the immobilization of TiO2 on Stober sílica surface. The N2 adsorption and desorption isotherms revealed that the synthesized composites are mesoporous materials with mean pore sizes between 3 and 4 nm with approximately 20% of porosity. The estimated band gap energies were estimated for the composites were higher than that for the standard oxide. The authors have also shown that the composites showed excellent photocatalytic performance. In short the authors have provided much details about their findings and their conclusions are well stated based on their research. I recommend to publish this article in the journal.

Reviewer 2 ·

Basic reporting

1. There several typos in the manuscript. I recommend the authors proofread the final manuscript.
2. In the introduction, the authors need to clearly establish the motivation behind the study, how the results from their study going to help the other researchers working in this area? This is missing in the current manuscript.
3. Also, comment on how the photocatalytic performance of the TiO2@SiO2 catalysts presented in this study compares with TiO2@SiO2 catalysts reported in the literature.

Experimental design

Nothing to report.

Validity of the findings

Nothing to report

Additional comments

1. In certain parts of the the description of the results is ambiguous. For instance, on page 13, lines 327-330, the description is not clear, when the authors simply mention "the Eg higher". What is the being compared with what? Is this high compared to literature? Clarify?
2. In the submitted manuscript, the color labels are either missing or not clear. I recommend adding legends in each figure. This will help readers to look at the sample names directly in the figure. Also, mark/label the peaks of interest in figures pertaining to XRD and Raman study results.

Reviewer 3 ·

Basic reporting

Basic reporting is good but can be improved

Experimental design

Authors should include few more experimental techniques and results to support the findings.

Validity of the findings

Authors should improve the "Discussion" section and can be explain their results in a detailed way to convince the data rather reporting their findings.

Additional comments

1. Authors should mark the peaks in the FT-IR spectra and label each spectrum as inset or should use better appropriate way to distinguish each spectrum (applicable to all the diagrams).
2. It should be the arbitrary unit (a.u.) in the Y axis of figure 1.
3. Please make sure the spellings are correct through out the text. E.g. “Diffractogram” – wrongly spelled through out the text.
4. Authors should bring in more experimental results in order to support the objective of the research work.
5. The manuscript could be clear and professionally written to meet the standards of the journal.

---

## Round 0.2 · Minor Revisions

I believe the reviewer's comments were considered and incorporated adequately. Hence, the manuscript can be accepted for publication.

---

## Round 0.3 · accepted · Accept

After the improvements made during the revision, the manuscript can now be accepted for publication.